# Examining the Heat Health Burden in Australia: A Rapid Review

Manoj Bhatta [1,*], Emma Field [2], Max Cass [3], Kerstin Zander [4], Steven Guthridge [1], Matt Brearley [5], Sonia Hines [6], Gavin Pereira [7], Darfiana Nur [8], Anne Chang [1], Gurmeet Singh [1], Stefan Trueck [9], Chi Truong [10], John Wakerman [1] and Supriya Mathew [1]

1   Menzies School of Health Research, Charles Darwin University, Casuarina 0810, Australia; supriya.mathew@menzies.edu.au (S.M.); steve.guthridge@menzies.edu.au (S.G.); anne.chang@menzies.edu.au (A.C.); gurmeet.singh@menzies.edu.au (G.S.); john.wakerman@menzies.edu.au (J.W.)
2   College of Health and Medicine, Australian National University, Canberra 2601, Australia; emma.field@anu.edu.au
3   Monash Health, Clayton 3168, Australia; max.cass@monashhealth.org
4   Northern Institute, Charles Darwin University, Casuarina 0810, Australia; kerstin.zander@cdu.edu.au
5   National Critical Care and Trauma Response Centre, Eaton 0810, Australia; matt@thermalhyperformance.com.au
6   Centre for Remote Health, Flinders University, Adelaide 5042, Australia; sonia.hines@flinders.edu.au
7   School of Public Health, Curtin University, Bentley 6102, Australia; gavin.f.pereira@curtin.edu.au
8   Department of Mathematics and Statistics, The University of Western Australia, Perth 6009, Australia; darfiana.nur@uwa.edu.au
9   Macquarie Business School, Macquarie University, Sydney 2109, Australia; stefan.trueck@mq.edu.au
10  Centre for Financial Risk, Macquarie University, Sydney 2109, Australia; chi.truong@mq.edu.au
*   Correspondence: manoj.bhatta@menzies.edu.au; Tel.: +61-08-89595220

**Abstract:** Extreme heat has been linked to increased mortality and morbidity across the globe. Increasing temperatures due to climatic change will place immense stress on healthcare systems. This review synthesises Australian literature that has examined the effect of hot weather and heatwaves on various health outcomes. Databases including Web of Science, PubMed and CINAHL were systematically searched for articles that quantitatively examined heat health effects for the Australian population. Relevant, peer-reviewed articles published between 2010 and 2023 were included. Two authors screened the abstracts. One researcher conducted the full article review and data extraction, while another researcher randomly reviewed 10% of the articles to validate decisions. Our rapid review found abundant literature indicating increased mortality and morbidity risks due to extreme temperature exposures. The effect of heat on mortality was found to be mostly immediate, with peaks in the risk of death observed on the day of exposure or the next day. Most studies in this review were concentrated on cities and mainly included health outcome data from temperate and subtropical climate zones. There was a dearth of studies that focused on tropical or arid climates and at-risk populations, including children, pregnant women, Indigenous people and rural and remote residents. The review highlights the need for more context-specific studies targeting vulnerable population groups, particularly residents of rural and remote Australia, as these regions substantially vary climatically and socio-demographically from urban Australia, and the heat health impacts are likely to be even more substantial.

**Keywords:** heatwaves; climate change; heat stress; hot weather; extreme events

## 1. Introduction

The frequency, intensity, and duration of extreme heat events have increased for most regions globally [1], including Australia, which experienced the warmest year on record in 2019 since 1910 [2]. Temperature increases are observed across all seasons, with both day

and night-time temperatures rising across the continent [3], posing significant health threats in the form of increased morbidity and mortality [4–8] and thus increasing demand on the healthcare system [9–11]. In Australia, it has been reported that hot days and heatwaves (HWs) together have resulted in more deaths than any other natural hazard, including bushfires, cyclones, earthquakes, floods and severe storms combined [12].

The average Australian surface temperature is projected to increase between 2.8 and 5.1 °C by 2090 [13]. Knowledge of how heat affects the diverse Australian population is essential for decision makers to implement adaptation strategies and early warning systems as the climate changes. It also assists in projecting health service utilisation rates and health and emergency service resource planning [14]. Given that vulnerability and susceptibility to heat-related health impacts vary between people and places [15], many Australian studies have examined the impacts of heat on various health aspects [16,17] and also on the economic burden of heat on health [18,19]. A recent Australian systematic review found that the use of different HW definitions and health outcome variables and methodologies meant that synthesising the actual impact of HWs on health service demand across Australia was complicated [20]. Another review found limited studies that examined HW-related demand for prehospital retrieval services in rural and remote Australia but highlighted that the absence of evidence does not mean that there is no heat-related demand for rural/remote prehospital medical services [21]. To date, there has not been a systematic analysis of heat health articles to identify gaps in knowledge related to the varying geographical, climatic and social contexts in Australia. This review seeks to address this gap.

The specific objectives of our review were to (1) understand the geographical variability of heat health research in Australia, (2) identify the range of heat exposure and health outcome variables used in heat health impact studies, and (3) determine the trends related to the impacts of heat on health across climate zones. The review findings have global relevance as Australia covers a range of climate zones (equatorial, tropical, subtropical, desert, grassland, temperate). An understanding of climate zone-specific heat-related health effects is useful to other regions with similar climate zones internationally.

## 2. Methods

Rapid reviews are an emerging approach to synthesising evidence in which elements of the systematic review method are streamlined to generate information within a short timeframe [22,23]. An adapted version of the rapid evidence synthesis process proposed by Khangura and colleagues [24] was adopted. Steps included systematic literature search, study screening and selection of studies, data extraction and synthesis.

A search strategy was designed to identify relevant English-language peer-reviewed journal articles that have quantitatively examined the impact of heat on health using secondary health outcome datasets for the Australian population. The search was conducted in three electronic databases—Web of Science, PubMed and CINAHL. Articles from 2010 to 2023 were included in this review. The final search was undertaken on the 18th of January 2023. The search terms were related to HWs and health and the geographical location of Australia. The complete list of search terms and databases is provided in Table S1. Articles were screened using the inclusion/exclusion criteria provided in Table 1.

**Table 1.** Inclusion and exclusion criteria for the rapid review.

| Included | Excluded |
| --- | --- |
| Full journal peer-reviewed articles published in English between 1 January 2010 and 31 December 2022. | Non-English language publications. |
| Quantitative studies conducted in various geographical locations in Australia. | Qualitative studies. |
| The exposure variables included various temperature variables/indices derived mostly from the Bureau of Meteorology weather data or hot weather events that occurred in Australia. | Conference proceedings and abstracts. |

**Table 1.** *Cont.*

| Included | Excluded |
|---|---|
| The health outcome variables were to be measured using pre-existing or routinely collected data, including but not limited to mortality data, health service utilisation data (hospital inpatient data, primary healthcare data, emergency department data), emergency services utilisation data (ambulance call-out data), pre-hospital demand, perinatal data and occupational injury/illness data. | • Studies examining only ambient temperature without linking it to secondary health outcome data.<br>• Studies that solely explored seasonal variation of various health outcome variables.<br>• Studies where primary data was collected (e.g., surveys). |

Two independent reviewers initially screened titles and abstracts. Reviewers met to reach a consensus regarding their inclusion/exclusion decisions. Full-text reviews and data extraction were conducted mainly by one author. A second author was involved in full-text reviews and data extraction for 10% of the included articles for validation purposes.

A total of 132 eligible articles were included in the final rapid review (Figure 1). For each study, the following data were extracted: Author, title, year of publication, exposure variables, health outcome variables, a summary of results, and study location(s). The location(s) in the included articles were further classified based on the Australian Bureau of Meteorology climate zone classifications [25] and the Australian Bureau of Statistics remoteness classifications [26].

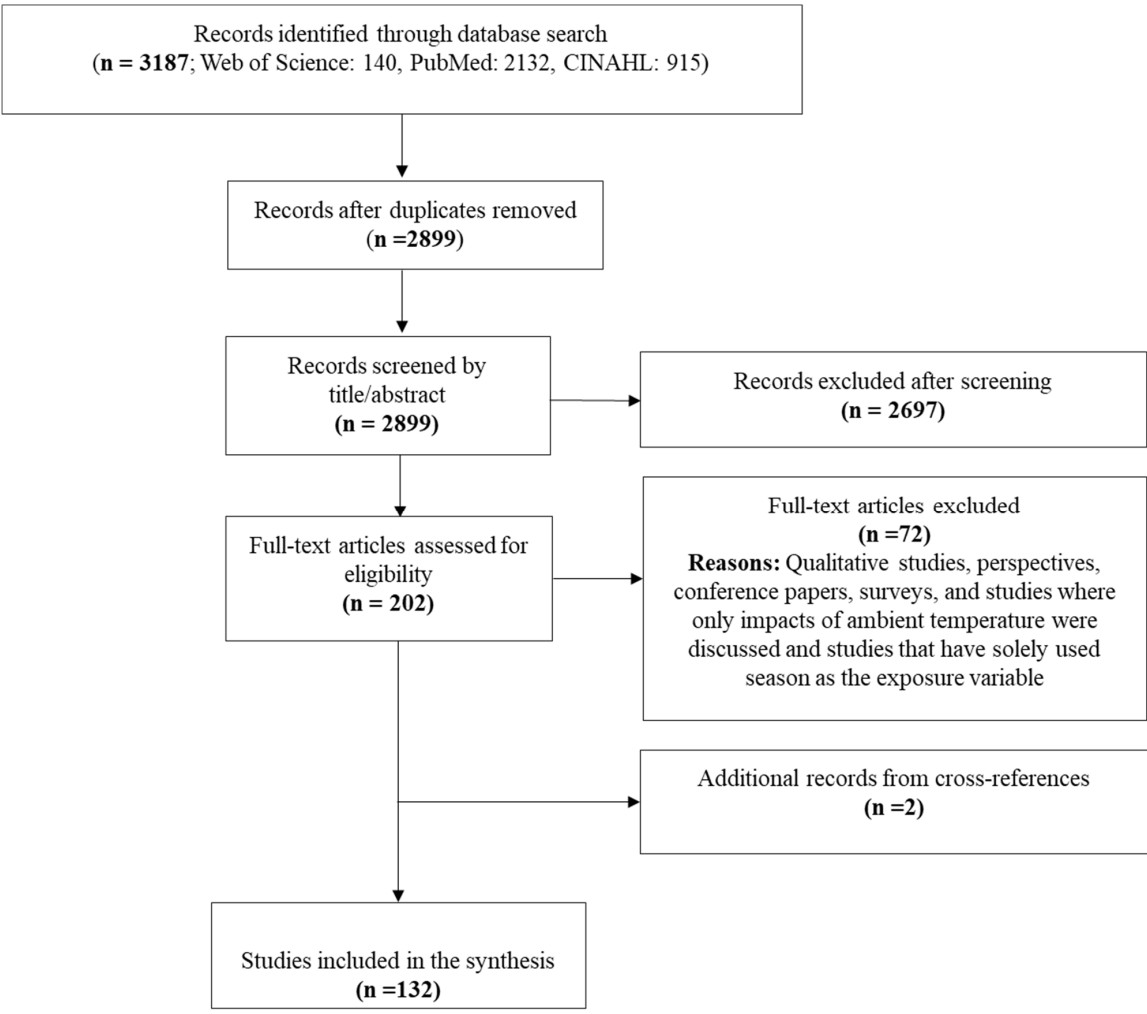

**Figure 1.** Flow diagram summarising rapid review search.

## 3. Results

### 3.1. Geographic Variability of Studies across Australian Jurisdictions

Two articles used temperature and health outcome data from across Australia [27,28]. The study by Longden (2019) [27] examined the impact of temperature on mortality across 548 local government areas, and the study by Qi and the team in 2015 [28] assessed associations between climate variability, unemployment and suicide rates in eight locations, including an inner regional area, an outer regional area, and six major cities across Australia.

Half of the articles (*n* = 66; 51%) included in this review were for studies that included temperature and health data from Queensland. Two of these were Australia-wide studies and 16 were multi-city studies. Almost all of the studies (*n* = 65) included Brisbane—the capital city of Queensland (Figure 2). Four articles covered data from across Queensland [29–32], three articles focused on other major cities of South East Queensland, including the Gold Coast and the Sunshine Coast [33–35], and four focused on regional towns, including Cairns, Townsville, Rockhampton, Mackay, Toowoomba, Mount Isa and Longreach [36–39].

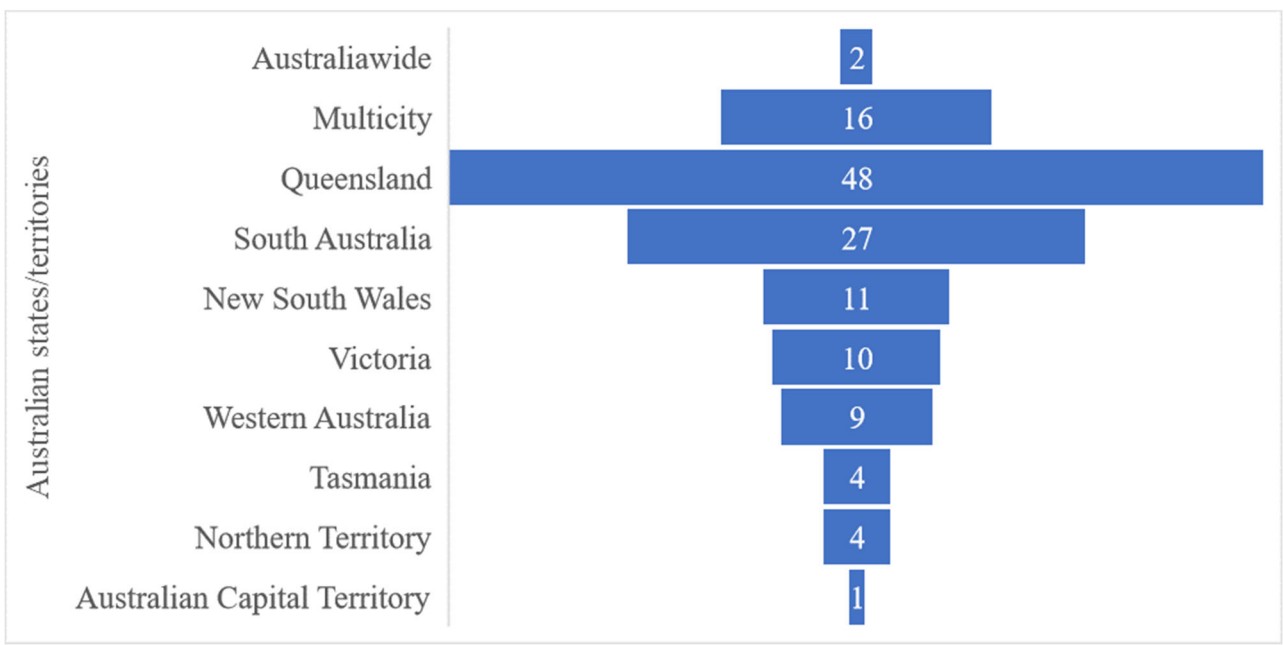

**Figure 2.** Geographic variability of the number of studies across Australian jurisdictions. Multi-jurisdictional studies were included in all relevant jurisdictions.

Of the 32 South Australia (including Australia-wide and multi-city studies; SA)-based studies (24% of all articles), almost all (*n* = 31) included data from its capital city, Adelaide, while one article covered the whole of SA, including inner regional areas and remote areas [40].

Twenty percent of all articles (*n* = 27, including Australia-wide and multi-city studies) were from New South Wales (NSW), which included Sydney, of which 14 were multi-city studies. Three studies used data beyond the Sydney region: one used data from the whole of NSW [41], and the other two articles included data from Illawarra, Gosford, Wyong, Newcastle and Wollongong [42,43].

There were 27 articles (20% of all the articles included in this review) that used data from Victoria (including articles that included multiple cities/whole of Australia). Studies were mainly Melbourne focused, with three exceptions: those that examined data for the whole of Victoria [44], inner and outer regional areas such as Bendigo, Wodonga, Latrobe Valley, Horsham, Hamilton, Lakes Entrance, Geelong, Shepparton, Ballarat and Mildura [45] and inner regional areas of South West Victoria [46].

There were 15 articles (11%, including Australia-wide and multi-city studies) in Western Australia (WA), predominantly focussed on Perth (*n* = 11), with two articles including the whole of WA [47,48].

Tasmania, the Northern Territory (NT), and the Australian Capital Territory (ACT) had the least number of studies (Figure 2). Tasmanian articles included data from the capital city Hobart [49], Launceston [50], Sorell Council in South East Tasmania [51], and the whole of Tasmania [52]. Of the four NT-related articles, two were NT wide [53,54], one focussed only on Alice Springs, a remote town [55] and another on Darwin, the main urban centre [56]. There was only one article covering the ACT [57].

*3.2. Temperature Exposure Variables Used to Examine Heat Health Effects*

Most studies used more than one exposure variable for the purpose of conducting sensitivity tests. Daily temperature variables (maximum temperature (Tmax), mean temperature (Tmean) and minimum temperature (Tmin), HWs and excess heat factor (EHF) were the main exposure variables studied (see Table S2), while a few studies used other temperature indices such as the universal thermal climate index [58,59], apparent temperature [43,60–64] and wet bulb globe temperature [60,61,64]. One study also used humidex as an exposure variable [64].

The goodness of fit of humidity indices compared to absolute temperature indices varied depending on the health outcomes studied. The absolute temperature was a better fit than the apparent temperature for mortality [65]. HW and temperature indices had the best fit for cardiovascular admissions, while humidity indices had the best fit for respiratory admissions, and combined heat-humidity indices had the best fit for renal admissions [60]. Some studies showed that irrespective of the temperature/humidity metric used, increased risks were observed for mortality and hospital admissions [43,62,66].

Heterogeneity in HW Definitions across Australian Literature and Its Effect on Impact Estimates

HW articles focussed on specific HW events or included HW definitions using varying temperature indicators, intensities and duration, and indices such as EHF and EHI [67] were used as exposure variables.

Heat health outcome estimates were found to depend on the HW definitions [68]. Changes to temperature exposure without time for acclimatisation have been linked to increased health service utilisation [69], and hence many recent Australian studies [40,41,47,50,52,60,70–78] have used EHF as an exposure variable, given that it accounts for people's acclimatisation to the local climate. A similar index, EHIaccl, has also been used as an exposure indicator in two studies [69,79]. EHF was used to study hospital health service utilisation in the main cities: Adelaide [40,70,71], Sydney [60–62], Perth [47,60,75], Brisbane [60,72] and Hobart [50]. EHF was also used to examine ambulance call-outs in SA [73,80,81], NSW [41] and WA [74]. Mortality studies were also conducted in NSW [41,62,79], WA [47,48,74], SA [40,82], Victoria [79], and Queensland [72] using EHF as the exposure variable.

Studies have also defined HWs as events when daily temperatures Tmax, Tmean, or Tmin are above a threshold (e.g., 90th/97th/99th percentile of whole year temperature data, summer data, or warm season data) for a number of consecutive days (2, 3, 4, etc.) [27,29,30,33,37,42,45,49,68,76,78,83–111]. Heat health effect estimates were also found to be sensitive to the temperature thresholds used as well as the duration of HWs [88] across the studies.

The use of specific temperature indicators also varied between the studies. A multi-city study [88] showed that Tmean was the best temperature indicator for HWs in Brisbane, while Tmax was found to be the best indicator for Melbourne. Most articles in this review used maximum daily temperatures for the HW definition (see Table S2), as it was considered a good representation of daytime temperatures. A study by Xu and Tong [88] observed that in Melbourne and Sydney, night-time relief did not reduce the health burden

caused by hot daytime temperatures. Three articles that examined the increased impact of HWs on outdoor workers in South Australia and Adelaide have used Tmax for the HW definitions [86,95,112]. On the other hand, some articles have also used Tmean (average daily Tmax and Tmin), describing it to better represent exposure to both day and night temperatures [76,88,93,94,97,99]. A couple of articles have compared the use of temperature indicators across various Australian locations and have found Tmax to be a better indicator for morbidity in NSW [43] and mortality in Melbourne [88]. Tmean was found to be a better indicator of mortality in Brisbane [88]. Exposure to high nighttime temperatures increased the risk of mortality in Sydney [88] and Perth [104].

Studies also used health-risk-based thresholds to define HWs. The thresholds are defined by exploring temperatures beyond which the health effects (e.g., mortality or morbidity) of HWs increase [76,78,98,107]. A multi-city study spanning Sydney, Brisbane and Melbourne showed that the relative risk of mortality started to increase around the 95th percentile of Tmean, increased sharply at the 97th percentile and had a much sharper increase at the 99th percentile during the warm season [76]. The mean temperature thresholds for mortality were 28.0 °C for Brisbane (97th percentile), 27.3 °C for Sydney (96.5th percentile) and 27.2 °C for Melbourne (96.5th percentile). Another study for Perth showed that Tmax and Tmin thresholds ranged between 34 °C, 36 °C and 20 °C, respectively [104]. In an Adelaide-based study, increased risks to heat-related mortality and morbidity were observed for Tmax and Tmin thresholds of 30 °C and 16 °C for mortality, 26 °C and 18 °C for ambulance call-outs, and 34 °C and 22 °C for heat-related emergency department (ED) presentations [78]. The Adelaide study did not show much variation in heat thresholds for individuals aged ≥ 65. Several articles also used the Australian Bureau of Meteorology definition of HWs for each state and territory [47,95,110], but sensitivity tests were also usually conducted by varying the duration of the HWs, temperature thresholds and temperature indicators.

### 3.3. Geographic Variability of Health Outcome Variables

More than half of the articles included mortality as at least one of the outcome variables. Forty-one articles explored all-cause mortality, while thirteen explored cause-specific mortality. The range of morbidity-related health outcome variables included hospital health service or emergency service utilisation, pre-hospital demand, perinatal outcome data and occupational injury/illness data (see Table S2). Only one study explored the impact of heat exposure using primary health care (PHC) utilisation data [113].

#### 3.3.1. Mortality

In one study, mortality was studied for the whole of Australia [27], where mortality risk was analysed by climate zones, capital cities, regional areas and socio-economic areas of advantage/disadvantage. Another Australia-wide study examined the impacts on suicide rates in eight Australian capital cities [28]. Various multi-city studies also examined heat-mortality associations [63,65,76,79,88,93,97,114–119].

In SA, mortality studies mainly focused on Adelaide [78,82,107,108,120,121] while one study examined the whole of SA [40]. In WA, mortality was specifically studied for Perth [69,104] and the whole of WA [47,48]. In NSW, mortality studies predominantly focused on Sydney [42,61,62,122,123] while a single study assessed the whole of NSW [41].

While several multi-city mortality studies included Melbourne, one study examined mortality using data from a major metropolitan facility (Alfred Hospital) in Melbourne [124], one study included ten regional Victorian towns [45], and another study examined deaths during the 2014 HW event in Victoria [44]. Almost all of the Queensland-based mortality studies were from Brisbane [68,72,83,87,98,106,125–134], except one based in South East Queensland [35].

From the articles reviewed, it appears that the effect of heat on mortality is immediate, with peaks in risk of death observed on the current day or at a one-day lag in Australia [88,93,97,98,105,106]. However, there were variations in the patterns of mortality

lag by city. Elevated risk of death was observed on lags of 0–3 days for Brisbane [93], 0–2 days for Melbourne and a one-day lag for Sydney [97]. The lag also differed by sex and gender for mortality, with females, particularly older females (≥75 years of age), showing an increased risk of mortality with 0–3 days lag in Brisbane and Melbourne [93,97].

### 3.3.2. Hospital, ED and Ambulance Call-Out Data

In SA, articles that used ED visits [70,71,110,135], ambulance call-outs [73,78,81,107–109,121], and hospital admissions [80,84,101,120,136] targeted Adelaide, the main city, except for one SA-wide study, which also covered remote and inner regional areas such as Kangaroo Island, Yorke and Lower Eyre, Adelaide Hills, and Murray Mallee [40]. In WA, some articles focussed on Perth [69,74,75,77,104], while a few articles used data from across WA [47,48].

In Queensland, the majority of articles were centred on Brisbane and analysed ED visits, ambulance call-outs and hospital admissions data. Meanwhile, a few Queensland-wide [29–31,37,137] and South East Queensland wide [33,35,131] studies focused on ED visits and hospital admissions data. A single study assessed ED visit data from regional and remote towns: Cairns, Mackay, Mount Isa, Rockhampton, Toowoomba and Townsville [36].

Only one NSW-wide study used ED presentations and ambulance call-out data [41]. All other NSW-based studies that examined hospital admissions [61,62,66,138,139], ED visits [122,140] and ambulance call-outs [122] concentrated on Sydney. One article also examined hospital admissions in Illawarra, Gosford-Wyong and Newcastle in addition to Sydney [43].

In Victoria, one state-wide study assessed ED presentations in nine Southwest Victorian hospitals [46]. Melbourne-focussed articles used ED visits [94,124], hospital admissions [111,124,141] and one at a sexual health centre presentation [113]. In Tasmania, ED visit data in Hobart and Launceston [49,50] and Tasmania-wide ambulance call-out data [52] were used to examine the heat health impacts. Hospital admission data from Darwin, Alice Springs, Gove, Katherine and Tennant Creek Hospitals [54,56,116] were used to examine heat health impacts in the NT. Only one study in ACT focused on ED heat-related presentations in Canberra [57].

### 3.3.3. Other Morbidity Indicators

Heat-related occupational injuries and illnesses were studied for Melbourne, Perth, Brisbane [85,142] and Adelaide [86,90,92,95,112,143] using workers' compensation claims data. One study [73] used work-related ambulance call-out data for Adelaide.

The impact on foodborne diseases such as salmonellosis and campylobacter was studied for Adelaide [91,144,145] and South East Queensland [34,39]. The impact was also measured by association with infectious diseases such as croup [140] in Sydney, dengue in Cairns and Townsville [38], influenza in Brisbane [146,147] and Melbourne and Sydney [147], cryptosporidiosis in Queensland [32] and Ross River virus cases in Southeastern Tasmania [51]. Four articles examined the impact of heat on perinatal outcomes, such as stillbirth and preterm birth in Brisbane [102,148–150] and one in Alice Springs [55].

### 3.4. *Impact of Hot Weather on Health by Climate Zones*

There are six major climate zones in Australia according to the Köppen climate zone classification—equatorial, tropical, subtropical, desert, grassland, and temperate [25]. A substantial number of the included studies were conducted exclusively in the temperate (*n* = 56) and subtropical (*n* = 40) climate zones. There were 32 articles that included multiple climate zones (see Table S2). Only a few articles specifically focused on grassland [55] and tropical Australia [56].

### 3.4.1. Hot Weather Impacts on Health in Australian Deserts and Grasslands

A study exclusively conducted in a Central Australian town found that hot weather was positively associated with an increase in preterm births [55]. Positive associations between hot weather and hospital admission rates for acute respiratory disease [56] and

cardiovascular diseases [54] were observed in the desert regions of the NT. In Queensland's grassland climate zone, hot weather was positively linked to ED visits for acute kidney injury [36] and cause-specific ED visits [37].

### 3.4.2. Hot Weather Impacts on Health in Tropical Australia

In tropical Australia, hot weather was positively associated with mortality [28], ED visits [36,37], hospital admission rates [53,54,56,60], and the number of *Salmonella* infection cases [39]. One study found no significant association with the number of dengue cases [38].

### 3.4.3. Hot Weather Impacts on Health in Subtropical Australia

Of a substantial number of articles based on subtropical climate zones (including multi-climate zone studies; see Table S2), most articles linked hot weather with an increase in mortality risks [28,35,63–65,68,72,76,79,83,87,88,93,98,106,114–116,119,125,127–134]. Positive associations were found with ED visits [33,36,37,99,100,137,151–153], hospital admission rates [35,60,68,72,83,87,89,96,98,105,106,154], preterm birth and stillbirth occurrences [102,148–150], ambulance call-outs [14,103,155], and other morbidity indicators including incidences of salmonellosis [34,39], incidences of influenza among paediatric patients [146,147], out-of-hospital cardiac arrest attended by paramedics [156], and occupational injury and illness [85]. However, two articles exhibited negative associations with mortality [117,118], and another one negatively associated hot weather with the number of paediatric seasonal influenza case presentations [146].

### 3.4.4. Hot Weather Impacts on Health in Temperate Australia

In temperate Australia, most of the articles linked hot weather positively with mortality [28,40,42,45,62,63,65,69,76,78,79,88,93,104,107,108,114–116,119–124], ED visits [36,37,40,49,50,57,69–71,75,78,80,94,104,108,110,121,122,124,135,140,141], hospital admission rates [40,43,60–62,66,69,78,84,101,108,111,120,121,124,135,136,138,139,157], ambulance call-outs [40,52,73,74,78,81,107–109,121,122] and other morbidity indicators such as the incidence of influenza B among paediatric patients [147], incidences of salmonellosis [91,145], daily *Campylobacter* cases [144], Ross River virus cases [51] and occupational injury and illness [85,86,90,92,95,112,142,143]. Negative associations were found against ED visits and total hospital admissions [46,104], whereas associations were not made in the two studies [101,113].

### 3.5. Population Subgroups

This review found that several population subgroups, including children, the elderly, Indigenous people, people with socio-economic disadvantage, pregnant women, remote/rural residents, workers and residents of different climate zones, were specifically studied in the articles (see Table S2). The literature, however, focused predominantly on metropolitan populations.

### 3.6. Currency of Datasets

Of the 132 articles reviewed, 67 articles used data that can be considered contemporary (defined as data derived from years that include the year 2010 or later years), while 64 articles used data extracted entirely from records predating 2010. The one remaining study, published in 2014, did not specify the period of health data used [140].

## 4. Discussion

Our rapid review found that most articles indicated increased heat-related mortality and morbidity risks across all climate zones in Australia. The health effects were sensitive to the definition of HW and the type of temperature exposure variables. The review has clearly identified gaps in evidence related to heat health effects on Australians living in rural and remote locations and tropical and desert regions. Jurisdictions such as the NT, despite having a high contribution of heat-related national disaster deaths [158], have limited information on how heat affects its residents. A quarter of the NT population

constitutes First Nations people who experience a lower life expectancy, poorer perinatal outcomes and a higher prevalence of cardiovascular, renal, respiratory, mental health and diabetes-related diseases [159]. High ambient temperatures have been linked to all these health outcomes [11], yet only a few impact studies have reported on the health effects of First Nation's status [160]. People living in remote Australia often do not have the necessary adaptation infrastructure (e.g., air conditioners or thermally comfortable homes) to limit exposure to extreme heat. For example, almost half of the NT population lives in rented houses or social housing, affecting their ability to make structural modifications in response to the weather [161]. Many remote residents also experience energy poverty (energy bills ≥ 10% of household income) and issues accessing a reliable power supply due to financial constraints [162], contributing to their heat exposure. The studies that included rural and remote areas [41,47] in NSW and WA demonstrated an increased risk of adverse health impacts from remoteness. The more disadvantaged and most remote locations had a higher risk for morbidity from HWs. This highlights the need to focus more on communities living in remote Australia and on population groups experiencing high socio-economic disadvantage. The lower number of articles in jurisdictions such as the NT and WA could be because of the presence of many sparsely populated remote/very remote communities that have limited access to healthcare facilities, particularly hospital services and the limited number of weather stations compared to urban centres [163].

Our review raises questions about the adequacy of health outcome datasets currently used in Australian literature. For many people living in remote and very remote locations [164], the first medical contact point is a remote clinic rather than a hospital [165]. This would mean minor to moderate heat-related symptoms for remote residents will be evident only through the analysis of PHC data and not hospital, ED, or medical retrieval data. We also argue that current heat-related health impacts are underestimated as PHC data have yet to be adequately analysed across Australia [166]. Not all heat-related effects will require hospital admissions, emergency visits, or ambulance call-outs [167]. In addition to PHC data, there are gaps in the use of health outcome datasets [18]. Studies conducted across jurisdictions can assist in understanding the differential impacts of heat on health in Australia. Studies utilising health data across Australia are limited by the difficulty in accessing health data from across jurisdictions. Ownership of hospital/ED data is held by individual state/territory health departments and private organisations with their own data requests and approval processes. Access to an Australia-wide PHC dataset is more complicated as PHC clinics include government-controlled, privately run and community-controlled clinics, which means Australia-wide studies focussed on PHC utilisation would need to obtain approval from several data owners.

Several studies used data from before 2010. Due to the inherent variability of meteorological data, increasing temperature thresholds and their evolving impact on the health outcomes of Australians, studies that aim to correlate changes in meteorological conditions with health outcomes must be updated regularly. Further research should be undertaken to ensure the currency of the findings of these older studies.

Current evidence is not adequate to understand the health effects on transient population groups. For example, most rural and remote locations in Australia rely on a short-term workforce, including fly-in-fly-out workers and short-term visa holders working on farms or in the hospitality industry [168], who need time to acclimatise to the local heat conditions [169]. Prolonged heat exposure by outdoor workers results in heat stress [170] and heat-related symptoms [171] with cumulative effects from successive work shifts [172]. Examining workers' compensation data and organisational health and safety databases across jurisdictions helps reveal the impact of such heat exposure on workers' health [18]. There is a deficit of studies that target populations susceptible to heat exposure, such as participants of sports and recreational activities [173] and tourists and short-term workers that have not been identified in routinely collected health outcome datasets identified by this review. Tourists visiting from the northern hemisphere during the Australian summer lack adequate heat acclimatisation and local experience, thereby increasing the risk of heat-

related health impacts. This effect is supported by the heat-related deaths of ten Northern Hemisphere tourists in the NT during the 2003–2018 period [158]. The review found no papers that included tourists or short-term visa holders as a sub-group for analysis. During health service encounters, detailed information about individuals may not be recorded, which makes it hard to identify and explore the effects of hot weather on such population groups. Targeted studies will be required to understand the heat health impacts on such at-risk groups.

While there is ample evidence from the southern parts of Australia (though urban focussed) that extreme heat is associated with increases in mortality, all-cause healthcare presentations, and specific cause healthcare presentations such as diabetes, respiratory illness, cardiovascular disease, and mental illness [11], this Southern Australia-derived evidence highlights the need to put in place heat health risk reduction measures for at-risk populations in Northern Australia. Due to the combined effect of heat and humidity, Northern Australia, the region above the tropic of Capricorn in Australia, presents a very different experience of heat compared to Southern Australia. High humidity levels significantly contribute to heat stress by impeding evaporative cooling [15], highlighting the requirement to include humidity-related exposure variables for the region. The Northern Australian population is also more susceptible to heat-related health risks due to the presence of people with poorer socio-economic conditions, pre-existing health disadvantages, and a transient population that may not have acclimatised to the local weather [53,174]. Particular attention should be given to the climate metric used to define heat in Northern Australia. EHF has been increasingly utilised and reported as a superior predictor of health service utilisation in hot and dry summer climates. Yet, the application of EHF to tropical regions has been questioned due to humidity not being accounted for in the index [15]. EHF that uses humidity-affected temperature indices could be used for tropical regions [175]. The advantage of using the EHF is that it factors in people's acclimatisation to the local climate and is effective in places where there is more heat variability. While this review found many studies using EHF as an exposure variable, no heat health impact studies used EHF in desert and grassland environments in Australia. A single temperature exposure variable may not suit all climate zones [60], so understanding the best temperature exposure variable is also a priority for heat health impact research.

Only a few studies have investigated the effects of maternal exposure to hot weather on birth outcomes, while there is a large international literature that reports mixed effects on pregnant women [176]. The maternal heat exposure–preterm birth link is important due to the life-long impact of complications, social and economic costs associated with preterm births and the high preterm birth rates [58,59] observed among First Nations populations in Australia (currently double the rates of the non-Indigenous population) [177].

The studies that investigated the impact of heat on foodborne diseases demonstrate increased risks. Changes in humidity and rainfall could also accompany extreme hot weather, changing the dynamics of human infectious diseases by impacting pathogens, vectors/hosts, or transmission routes and thus requiring further research attention. While daytime temperatures have been predominantly examined in the literature, elevated overnight temperatures can also affect people's health through prolonged periods of heat exposure. Residents of densely populated regions are particularly at risk of high night-time temperatures due to the urban heat island effect [178,179]. Similarly, socio-economically disadvantaged people (e.g., homeless people) who do not have the necessary protective infrastructure will be more affected by HWs when there are high night-time temperatures. Public spaces such as shopping centres, swimming pools, or art centres often used as heat respites are closed during the night, which affects people's ability to seek heat refuge.

Our review did not include a meta-analysis comparing mortality or morbidity risk estimates across geographical locations, as health outcome variables and definitions varied between articles. In terms of understanding the impacts, there are also several seasonal variables that can be linked to heat-related health effects, such as school/university attendance/withdrawals [180], alcohol/drug consumption [181] and criminal activity [182,183],

that could be used to study heat health impacts but have not been the focus of this review. Such routinely collected datasets can also be linked to various health outcome variables to further explore heat health impacts.

## 5. Conclusions

A positive association between extreme heat and adverse health outcomes is generally reflected in Australian studies, with our review also documenting the geographical, climatic and social context of this field of research. Our review indicates an absence of population-level heat health impact studies for rural and remote areas and tropical and desert regions of Australia. The unequal geographic distribution of studies and the lack of PHC data analysis underestimate the actual impacts on the Australian population. Routinely collected health outcome datasets are limited by the fact that they do not capture heat health effects on transient populations and people who are involved in outdoor activities. Policymakers and key stakeholders thus need more evidence of the actual impacts to develop context-specific adaptation strategies or heat health alert systems.

**Supplementary Materials:** The following supporting information can be downloaded at: https: //www.mdpi.com/article/10.3390/cli11120246/s1, Table S1: Search terms used in the rapid review; Table S2: Data extracted from the included articles.

**Author Contributions:** S.M. conceptualized the study, E.F., M.C., S.M. and M.B. (Manoj Bhatta) designed the study; S.M. and M.B. (Manoj Bhatta) conducted the initial screening.; M.B. (Manoj Bhatta) and S.M. led the drafting of the manuscript; K.Z., S.G., M.B. (Matt Brearley), S.H., G.P., D.N., A.C., G.S., S.T., C.T., and J.W. reviewed a subset of studies at the data extraction stage, revised the manuscript and agreed to the published version of the manuscript. All authors have read and agreed to the published version of the manuscript.

**Funding:** The review was funded by Spinifex Network. M.B. (Manoj Bhatta) is supported by the Healthy Environments and Lives (HEAL) Network Fellowship.

**Data Availability Statement:** The full search terms/strategy used in this rapid review have been published in the Supplementary Information. We searched the following databases: Web of Science https://www.webofscience.com/wos/woscc/basic-search, accessed on 18 January 2023), PubMed (https://pubmed.ncbi.nlm.nih.gov/, accessed on 18 January 2023), and (https://web.s.ebscohost. com/ehost/search/basic?, accessed on 18 January 2023). Details on data extraction with citations of included studies are provided in Table S2.

**Conflicts of Interest:** The author Max Cass was employed by the Monash Health. The author Matt Brearley is the managing director of Thermal Hyperformance Pty Ltd., which provides heat stress management services to maximise the health, safety, and performance of heat-exposed workers/athletes within industrial, government, and sporting organisations. The results reported within this paper do not materially alter the nature of this work. The remaining authors declare that the research was conducted in the absence of any commercial or financial relationships that could be construed as a potential conflict of interest.

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
