# Peer review of "Examining the Heat Health Burden in Australia: A Rapid Review"

_climate, doi:10.3390/cli11120246_

Round 1
Reviewer 1 Report
Comments and Suggestions for Authors
This paper describes a thorough review of published studies of relationships between heat and health in Australia. I have a small number of comments below to be addressed.
The body of the paper refers to Appendices 1 and 2, but the information is in Tables S1 and S2. Please update the paper to refer to the tables.
Do the authors know why a large proportion of the studies selected focused on Queensland? Is this due to location of health-focused research institutes, easier access to health data, for example?
Section 3.2 lists a number of temperature exposure variables but only a small number are discussed in section 3.2.1. What thresholds or other approach were used with the combined temperature-humidity indices? For example, for wet bulb globe temperature, were the thresholds for US military personnel used? Was there any indication that the combined temperature-humidity indices performed differently to indices based purely on temperature? Similarly, were the combined indices only used in locations where high temperatures and high humidity coincide (in Northern Australia)? It would also be good if the discussion included a summary of the behaviour of the combined heat-humidity indices on health metrics and whether they showed the same or different results if just temperature was used.
Specific comments:
L109, It would be more correct to say: Almost all the studies ...
L147-8: correct sentence to “… apparent temperature, wet bulb globe temperature and heat index …”. One study also the humidex which could be mentioned here. Please include the references which used each of these combined indices in this sentence.
L220: while a single study …
L243, replace "aimed at" with “focused on”. L244, remove word “only” and split into two sentences:
… hospital admissions data. A single study assessed ED …
L254, what are clinical presentation data?
L300, negatively associated hot weather with the
L308, interesting that one study found a positive association between hot weather and the incidence of influenza in paediatric patients. In the UK, influenza is much more common in later autumn and winter when air temperatures are cooler.
L312, whereas associations were not calculated in two studies”. Do the authors mean no association was found, or the study did not attempt to find an association?
L314, This review found that …
L323, needed to stipulate the years – does this phrase mean the study did not specify the years of data used?
L326, across all climate zones
L327: Could drop the last words “used in the articles” from the end of the sentence.
L368, would need to obtain approval
L399-415, there is no mention of the studies using apparent temperature and other combined heat-humidity metrics.
L414-415, do the authors mean that EHF was not used in studies of heat impacts on health in desert and grassland environments?
L173-185 and L425-428, why are these lines in bold type?
Comments on the Quality of English LanguageNone
Reviewer 2 Report
Comments and Suggestions for Authors
Abstract
It is not clear why rural people would somehow react differently to heat compared to more urban populations. The lack of specific evidence might mean that we should rely on the evidence that is available for urban populations, rather than just conclude we do not have enough information to provide any policy guidance at all (although this point seems to be made later on line 61). This abstract seems to assume that context-specific data are needed to establish any policy guidance, which does not seem to be consistent in the paper.
The abstract seems to not have many specific findings. For example, this could be included: “From the articles reviewed, it appears that the effect of heat on mortality is immediate, with peaks in risk of death observed on the current day or at one day lag.” From line 227.
The abstract could also include some of the specific health effects identified.
Line 72. Perhaps the methods section should outline which temperature metrics were included. Was humidity not considered?
Line 293. Is it necessary to list all the specific references? This seems too long and not particularly helpful to the reader.
Line 342. It wasn’t the small number of studies that demonstrated the increased risk, it was the studies themselves. This sentence should be reworded.
Line 397. It is not clear why the authors have deemed these studies to be “insufficient.” Some of the risk populations have been identified in this paper, and it is doubtful the authors really mean that we cannot issue heat alert mechanisms at all, or that no alert mechanisms are somehow better than alert mechanisms adapted from other studies/regions. It is certainly appropriate to identify research gaps, but more problematic to suggest that no policy action can be developed at all. This seems to be the conclusion to the article as well. The default is to take no policy action, and this paper may inadvertently lend support to this tragic default situation.
Line 425. Is there some reason this section is in bold type?
Comments on the Quality of English Languagenone
Reviewer 3 Report
Comments and Suggestions for Authors
There are formatting errors and I am not sure why the text in lines 171 - 186 and lines 425 - 428 are in bold and larger font?
The references need to be reformatted.
This is a very useful review paper on a topic that is increasingly important in a warming climate.
